# Role of Polysaccharides from Marine Seaweed as Feed Additives for Methane Mitigation in Ruminants: A Critical Review

**DOI:** 10.3390/polym15153153

**Published:** 2023-07-25

**Authors:** Kit-Leong Cheong, Yiyu Zhang, Zhuoting Li, Tongtong Li, Yiqing Ou, Jiayi Shen, Saiyi Zhong, Karsoon Tan

**Affiliations:** 1Guangdong Provincial Key Laboratory of Aquatic Product Processing and Safety, Guangdong Province Engineering Laboratory for Marine Biological Products, Guangdong Provincial Engineering Technology Research Center of Seafood, Guangdong Provincial Science and Technology Innovation Center for Subtropical Fruit and Vegetable Processing, College of Food Science and Technology, Guangdong Ocean University, Zhanjiang 524088, China; klcheong@gdou.edu.cn (K.-L.C.);; 2Guangxi Key Laboratory of Beibu Gulf Biodiversity Conservation, Beibu Gulf University, Qinzhou 535000, China

**Keywords:** marine seaweed, polysaccharides, feed additives, ruminants, methane mitigation

## Abstract

Given the increasing concerns regarding greenhouse gas emissions associated with livestock production, the need to discover effective strategies to mitigate methane production in ruminants is clear. Marine algal polysaccharides have emerged as a promising research avenue because of their abundance and sustainability. Polysaccharides, such as alginate, laminaran, and fucoidan, which are extracted from marine seaweeds, have demonstrated the potential to reduce methane emissions by influencing the microbial populations in the rumen. This comprehensive review extensively examines the available literature and considers the effectiveness, challenges, and prospects of using marine seaweed polysaccharides as feed additives. The findings emphasise that marine algal polysaccharides can modulate rumen fermentation, promote the growth of beneficial microorganisms, and inhibit methanogenic archaea, ultimately leading to decreases in methane emissions. However, we must understand the long-term effects and address the obstacles to practical implementation. Further research is warranted to optimise dosage levels, evaluate potential effects on animal health, and assess economic feasibility. This critical review provides insights for researchers, policymakers, and industry stakeholders dedicated to advancing sustainable livestock production and methane mitigation.

## 1. Introduction

Global warming is a major environmental challenge that poses serious threats to the well-being of our planet and its inhabitants. This warming is mainly driven by an increase in atmospheric greenhouse gas concentrations, which trap heat in the Earth’s atmosphere and lead to rising temperatures. Methane, a potent contributor to global warming, has warming potential approximately 28 times greater than that of CO_2_ over a 100-year timescale [1]. Methane emissions result from both natural and human activities, including organic matter decomposition in wetlands, rice cultivation, landfilling, and the digestive processes of ruminants such as cows, sheep, and goats [2]. According to the Intergovernmental Panel on Climate Change, approximately 14.5% of global greenhouse gas emissions can be attributed to the agricultural sector, and enteric fermentation (i.e., methane emissions from livestock) is responsible for approximately 40% of these emissions [3]. As the global demand for meat and dairy products is expected to increase in the coming years, the problem of methane emissions from ruminants will intensify, making it imperative to reduce these emissions for environmental and economic reasons [4]. A range of strategies have been proposed, including dietary modifications, genetic selection, and improved herd management practices.

Methane emissions from ruminants result from enteric fermentation [5], which occurs in the digestive systems of animals, specifically in the rumen, a vast fermentation chamber that harbours a large number of microorganisms [6]. These microorganisms play a pivotal role in enabling ruminants to digest complex plant materials such as cellulose and hemicellulose, which cannot be broken down by monogastric animals such as pigs [7]. During enteric fermentation, the microorganisms in the rumen decompose feed materials and generate methane as a byproduct. Methane is released into the atmosphere via belching, contributing to increases in the concentrations of greenhouse gases and exacerbating warming global temperatures (Figure 1) [8]. Apart from the environmental impact, reducing methane emissions is crucial for the economic sustainability of animal agriculture, as numerous countries have established targets for the reduction of greenhouse gas emissions to meet international agreements and address climate change [9]. Additionally, reducing methane emissions from ruminants may lead to substantial improvements in animal health and productivity. Methane production results in energy depletion in animals, as it represents lost potential energy that could be used for growth or milk production [10]. By reducing methane emissions, more energy can be made available for production purposes, which may result in increased efficiency and profitability in animal agriculture.

Macroalgae, popularly known as seaweed, are large multicellular algae primarily found in marine habitats. By contrast, microalgae are single-celled algae that inhabit both marine and freshwater environments [11]. Both macroalgae and microalgae contain various intricate polysaccharides, such as laminaran [12], fucoidan [13], alginate [14], carrageenan [15], and porphyran [16], which exhibit several biological properties, including immune modulation and antioxidant, antiviral, prebiotic, and antimicrobial effects [17]. Marine algal polysaccharides (MAPs) mitigate the risk of inflammatory disorders in ruminants, improve food digestion, and augment nutrient absorption, thereby decreasing the probability of pathogen proliferation in the digestive system and enhancing overall animal health [18,19].

As shown in Table 1, some feed additives have been used to reduce methane production. As feed supplements, MAPs are generally deemed ecologically sound and sustainable as they are sourced from renewable sources and do not pose the same hazards or create the same concerns as other feed additives, such as antibiotics. These compounds hold promise in modifying the microbial populations in the rumen and/or inhibiting specific enzymatic pathways involved in methane production, so they are plausible candidates for the reduction of methane emissions from ruminants. Within this context, in this comprehensive review, our aim is to conduct an extensive examination of the role of MAPs as feed additives in minimising methane emissions in ruminants. This review is structured into three key components, each of which is crucial in understanding the potential of MAPs. First, we examine the effects of MAPs on ruminal microbial populations. By analysing existing research, we describe how MAPs may influence the composition and activity of rumen microorganisms, which are pivotal in methane production. The second focus is the effects of MAPs and volatile fatty acids (VFAs) on rumen fermentation and methane production. Understanding the interactions between MAPs and VFAs provides valuable insights into their ability to promote efficient fermentation pathways, leading to reduced methane emissions. Finally, we examine the antimicrobial activity of polysaccharides and their related mechanisms. Investigating these mechanisms is essential not only to optimise the effectiveness of MAPs but also to ensure the overall health of ruminants. This review highlights the importance of collaborative efforts between researchers and industries to facilitate the successful integration of MAPs into ruminant diets as a viable methane mitigation strategy. 

## 2. Marine Seaweed Polysaccharides

The various types of macroalgae include red (Rhodophyta), brown (Phaeophyceae), and green (Chlorophyta) algae [32]. MAPs are complex carbohydrates with high molecular weights that are derived from different seaweed species. Polysaccharides are naturally occurring compounds that are abundant in marine environments; they possess chemical structures and compositions that widely differ from those of terrestrial plants [33,34,35]. From a chemical perspective, MAPs typically consist of repeating monosaccharide residues such as glucose, mannose, galactose, fucose, rhamnose, mannuronic acid, guluronic acid, glucuronic acid, and iduronic acid. The arrangement and bonding patterns of these sugar units yield various polysaccharides, each contributing to their properties and diverse applications. MAPs exhibit remarkable characteristics, including biocompatibility, biodegradability, and low toxicity, so they are highly attractive for use in livestock, biomedical research, cosmetics, food production, and agriculture.

Red seaweeds, belonging to the phylum Rhodophyta, are an abundant source of a variety of polysaccharides, including carrageenans, agar, agarose, agaropectin, and porphyran [36]. Carrageenans are sulphated polysaccharides, and their classification into κ-, ι-, and λ-carrageenan is based on their sulphate content. κ-carrageenan is composed of alternating units of β-D-galactose and 3,6-anhydro-D-galactose, whereas ι-carrageenan consists of repeating units of α-D-galactose-4-sulphate and 3,6-anhydro-D-galactose [37]. λ-carrageenan primarily consists of disaccharide units of β-D-galactose-6-sulphate [38] and is a complex mixture of agarose and agaropectin. Agarose is a linear polysaccharide composed of repeating units of β-D-galactose and 3,6-anhydro-L-galactose, whereas agaropectin is a branched polysaccharide containing additional monosaccharides such as xylose and sulphate groups [39,40]. Porphyran typically comprises alternating units of β-D-galactose-4-sulphate and 3,6-anhydro-L-galactose [41].

Brown seaweeds encompass a diverse range of marine algae and are abundant in various polysaccharides, with varying compositions among species. Fucoidan, laminaran, and alginate are the most prevalent polysaccharides found in brown seaweed. Fucoidan, a sulphated polysaccharide, is primarily composed of fucose as the main monosaccharide, along with other monosaccharides, such as galactose, xylose, and mannose [13]. The most prominent linkage type in fucoidan is the α-(1 → 3) linkage, connecting fucose units through an α glycosidic bond at positions C1 and C3. Additionally, fucoidans can contain other linkages, including -(1 → 4), -(1 → 2), and -(1 → 6) linkages. Alginate mainly consists of β-D-mannuronic acid and α-L-guluronic acid residues, with the primary linkage being the 1,4-glycosidic bond [17]. Laminaran, a β-glucan polysaccharide, predominantly comprises glucose units linked by β-(1,3)-glycosidic bonds, accompanied by β-(1,6) branching [12].

Green seaweeds are a valuable source of polysaccharides, with ulvan being the most commonly reported sulphated polysaccharide found in these seaweeds. Ulvan has a complex structure and is composed of various monosaccharides, including xylose, rhamnose, glucuronic acid, and iduronic acid [42]. The backbone of ulvan primarily consists of interconnected glucuronic acid and iduronic acid units, which are linked together by β-(1 → 4) glycosidic bonds. The sulphate groups attached at different positions along the backbone contribute to the overall negative charge of ulvans [43].

The inclusion of MAPs in ruminant diets generally has a positive or neutral effect on meat quality attributes. In some cases, MAPs have been associated with increased meat quality and meat tenderness [44,45], which can enhance consumer satisfaction. Similar to meat quality, the inclusion of MAPs does not change milk quality or composition [46]. Improvements in milk quality parameters, such as increased concentrations of beneficial fatty acids and antioxidants, have also been reported with the inclusion of MAPs [47,48].

## 3. Effects of MAPs on Rumen Microbial Populations

The rumen microbiota is a complex and diverse community of microorganisms, including bacteria, protozoa, and fungi. Among these microorganisms, bacteria dominate the digestive tracts of ruminants, with cell counts ranging from approximately 10^10^ to 10^11^ cells/mL and encompassing more than 200 species [49]. Bacteria play pivotal roles in the degradation and fermentation of polymeric carbohydrates in animal diets. Fibrolytic bacteria, such as *Fibrobacter succinogenes* and *Ruminococcus flavefaciens*, specialise in breaking down cellulose and hemicellulose polysaccharides [50]. Additionally, amylolytic and lactate-using bacteria contribute to the breakdown of starches and sugars, ensuring the efficient use of energy sources within the rumen [51,52]. Dietary complexity is positively associated with microbial diversity. The intricate interactions among these bacterial populations produce a wide range of metabolic activities that ultimately promote nutrient acquisition and support overall animal health.

MAPs have the capacity to affect both the structure and composition of rumen microbial communities. Moreover, MAPs can enhance the efficiency of microbial fermentation in the rumen. *Prevotella*, *Butyrivibrio*, and *Ruminococcus* are the main bacterial species in the rumen, and alterations in the host diet can influence the overall community structure [53,54]. A previous study involving North Ronaldsay sheep isolated *Prevotella* spp. and *Clostridium butyricum* from the rumen microbiota. This discovery highlighted the remarkable ability of these microorganisms to break down various brown seaweed polysaccharides, including fucoidan, laminaran, alginate, and carboxymethylcellulose [55]. Chitosan supplements with varying molecular weights of 1, 3, 5, 50, and 200 kDa were used in this study. These results indicated that chitosan with a molecular weight of 3 kDa is promising for the mitigation of methane production as it modulates the composition of the bacterial community. Specifically, it encourages the substitution of fibre-degrading bacteria (*Firmicutes* and *Fibrobacteres*) with amylolytic microbial species (*Bacteroidetes* and *Proteobacteria*) [56]. According to Zanferari et al., the incorporation of chitosan into the diets of dairy cows in the absence of lipid supplementation resulted in a decline in bacterial species such as *Butyrivibrio* and *Butyrivibrio proteoclasticus*, which are known to be involved in rumen biohydrogenation. This decrease in the bacterial population correlated with a reduction in milk yields. However, the addition of chitosan led to elevated levels of unsaturated lipids and cis-9, trans-11-conjugated linoleic acid in milk [57]. 

Methane production in the rumen is primarily attributed to the metabolic activity of methanogenic archaea, which are specialised microorganisms that generate methane as a byproduct. Notable examples of the methanogenic archaea found in the rumen include *Methanobrevibacter smithii*, *Methanosphaera stadtmanae*, *Methanomicrobium mobile*, and *Methanosarcina* spp. (Figure 2) [58,59]. These archaea use H_2_, CO_2_, and methanol for methane synthesis [60]. However, specific bacterial species within the rumen contribute to the fermentation process by providing substrates that support methanogenesis [61]. Although these bacteria are not directly involved in methane production, they play a crucial role in establishing favourable conditions for methanogenic activity. The incorporation of MAPs into ruminant diets can induce shifts in the relative abundance of particular bacterial taxa. Consequently, these shifts often reduce the population of methanogenic archaea. In a previous study, the effectiveness of incorporating *Asparagopsis taxiformis*, a red alga containing natural compounds that selectively inhibit specific enzymes in methanogenic archaea, into the diets of cattle and sheep was investigated [62]. Supplementation with brown algae extracts markedly influenced the abundance of cellulolytic bacteria, including *Ruminococcus albus*, *Ruminococcus flavefaciens*, and *Fibrobacter succinogenes*, as well as methanogenic archaea and ciliate-associated methanogens [63].

In conclusion, the rumen microbiota possess an extensive repertoire of enzymes that can hydrolyse MAPs into simpler sugars and other fermentation intermediates, creating a rich substrate pool for their own growth and metabolic activities. These interactions between MAPs and the rumen microbiota are vital in methane mitigation in ruminants. In addition, MAPs can modulate the rumen microbial community, favouring the growth in bacteria associated with reduced methane production and the suppression of methanogenic archaea populations. The specific mechanisms through which MAPs modulate microbial diversity and alter fermentation pathways may vary depending on the characteristics of the MAPs and the composition of the rumen microbiota in different animal species or individuals. Therefore, further investigation is required to elucidate the detailed mechanisms underlying these interactions and their impact on methane mitigation. Understanding these mechanisms will provide valuable insights into the potential of MAPs as effective feed additives for the mitigation of methane emissions from ruminant livestock, thereby reducing their environmental impact and promoting sustainable animal agriculture.

## 4. Effect of MAPs and VFAs in Rumen Fermentation and Methane Production

In the field of ruminant nutrition, an essential component playing a pivotal role in rumen health and overall productivity is VFAs. VFAs are the main end products of microbial fermentation in the rumens of ruminants. They are primarily produced via the anaerobic breakdown of carbohydrates by ruminal microorganisms. These acids include C2 to C6 carboxylic acids, including acetic, propionic, butyric, isobutyric, valeric, isovaleric, and caproic acids [64,65]. These VFAs act as energy sources for host animals and are essential for rumen fermentation and digestion. Fibre digestion in ruminants is facilitated by the interplay between VFAs and the rumen microbial population [66]. The microorganisms involved in VFA production are summarised in Table 2.

VFAs are vital components of ruminal ecosystems. A balance among VFAs is essential for optimal ruminal function and animal performance. The generation of VFAs in the rumen is closely linked to methane production [75]. The proper allocation of VFAs for different physiological processes ensures efficient energy use and supports growth, reproduction, milk production, and methane emissions [76]. Different factors, such as the diet composition, rumen microbial population, rumen pH, and management practices, can influence the production and composition of VFAs and methane emissions [77]. Understanding these factors allows nutritionists and producers to formulate diets and management strategies that promote the production of VFAs that are favourable for rumen health and animal productivity.

Methanogens use a diverse range of substrates during methane production, including formate, hydrogen, methanol, butanol, 2-propanol, 2-butanol, propanol, dimethyl sulphide, dimethylamine, and trimethylamine [78]. Anaerobic digesters predominantly harbour hydrogenotrophic methanogens, indicating that hydrogen acts as the primary substrate for methane generation [79]. However, competition for substrates occurs between VFAs and methanogens because higher concentrations of VFAs can compete with methanogens for available hydrogen [80]. MAPs generally either have no marked effect on or decrease the levels of VFAs in the rumen. For instance, when 4% brown seaweed byproducts were incorporated as feed additives, only minimal alterations in volatile fatty acid concentrations were observed after 24 h [81]. The inclusion of *Sargassum horneri* did not influence the overall production of VFAs, whereas the 4% incorporation of *Ulva* sp. resulted in a decline in total rumen VFA production. Notably, both marine algal species led to a reduction in the methane content in the rumen [82]. When *Laminaria ochroleuca*, *Gigartina* sp., and *Gracilaria vermiculophylla* were combined with corn silage, a negative effect on total VFA production was evident, which was accompanied by a decline in methane production [30]. 

The effects of marine algae on the content and composition of VFAs vary depending on the substrate used. The breakdown of intricate polysaccharides derived from marine algae and the subsequent generation of fatty acids rely on the involvement of diverse bacterial species [83]. Notably, the phylum Bacteroidetes is recognised for its wide range of carbohydrate-active enzymes (CAZymes), which are responsible for this process. CAZymes can be categorised as glycoside hydrolases (GHs), polysaccharide lyases (PLs), and carbohydrate esterases (CEs) based on their distinct enzymatic catalytic mechanisms. These CAZymes specifically target bonds within MAPs, leading to their cleavage into smaller sugar units that can then undergo further metabolism. For instance, the CAZyme-encoding genes associated with fucoidan degradation and the breakdown of fucoidan linkages are found in the families CE4, GH29, GH107, S1_17, and S1_25 [13,84]. Kalyani et al. discovered mrbExg5, an enzyme that demonstrates exo-β-1,3-glucanase activity toward β-1,3-linked glucooligosaccharides and laminaran. This glycoside bears a structural resemblance to a member of the GH5_44 family, which is prominently present in *Pseudobutyrivibrio* sp. ACV-2 is an isolate derived from the rumen of cows [85]. The presence of abundant VFAs reduces the accessibility of hydrogen to methanogens, consequently hindering their activity and ultimately leading to a decline in methane production.

In vitro fermentation and in vivo studies have demonstrated the multiple beneficial effects of VFAs on host health. For instance, the use of porphyran and its partially acid-hydrolysed derivatives derived from *Porphyra haitanensis* can increase the concentrations of acetate, propionate, isobutyrate, butyrate, isovalerate, and valerate in the rumen [86,87]. Acetate is a key VFA generated during the breakdown of fibrous materials in the rumen. Their primary function is to provide substantial energy to ruminants, thereby fulfilling their energy needs. Acetate is an easily accessible energy substrate for animals that facilitates the maintenance of essential physiological processes [88]. However, acetate, as a substrate for methanogenesis, contributes to elevated methane emissions [89]. Propionates are vital VFAs that play pivotal roles in ruminant nutrition and energy metabolism. They serve as key precursors for gluconeogenesis, a fundamental process in glucose synthesis [90]. Glucose is essential for various metabolic functions, including milk production in dairy cows [91]. Higher propionate levels are associated with enhanced milk yields, emphasising their importance as VFAs in lactating animals. Additionally, propionate production is negatively correlated with methane emissions because it competes with methanogens for available hydrogen [92]. Choi et al. examined the effects of incorporating dried *Sargassum fusiforme* on ruminal fermentation in vitro. The experiment involved testing four different doses of *Sargassum fusiforme* (1%, 3%, 5%, and 10% of the total ratio). The findings indicated that supplementation with *Sargassum fusiforme* resulted in elevated propionate production with a simultaneous reduction in methane production [93]. Therefore, increasing propionate production relative to acetate production has the potential to mitigate methane emissions in ruminants. Although butyrate is produced in smaller quantities than acetate and propionate, its importance remains: it acts as a valuable energy source for the rumen epithelium and contributes to rumen health and integrity [94]. Furthermore, butyrate actively participates in microbial protein synthesis, facilitating the production of essential amino acids that are necessary for the overall growth and development of animals [95].

VFAs also provide the benefit of lowering the fermentation pH to below 6.0, thereby inhibiting the proliferation of methanogenic microorganisms. The inhibitory growth of the ruminal methanogen *Methanobrevibacter ruminantium* was demonstrated when suspended with lauric acid and myristic acid in low-pH conditions (approximately pH 5–6). The results showed that the decline in methane formation may have been related to the decreased survival of *Methanobrevibacter ruminantium* via increased ATP efflux, potassium leakage, and an increasing degree of protonation [96]. VFAs may have direct or indirect toxic effects on protozoa-associated methanogens, which are the microorganisms responsible for methane production. Macroalgae and their secondary metabolites are effective in reducing methane production based on in vitro results. For example, *Asparagopsis taxiformis* reduces methane production by suppressing methanogenesis [97]. 

In summary, understanding the interplay between MAPs, VFAs, rumen fermentation, and methane emissions is crucial in developing sustainable strategies to reduce the environmental footprint of ruminant livestock. By manipulating the production and use of VFAs through dietary interventions, we can potentially reduce methane emissions without compromising animal health or productivity. Future research should focus on further elucidating the mechanisms by which VFAs influence methane production and exploring novel dietary strategies and additives that can optimise VFA profiles and mitigate methane emissions.

## 5. Antimicrobial Activity of Polysaccharides and Related Mechanisms

Adding antibiotics such as ionophores (e.g., monensin and lasalocid) to ruminant diets can improve propionate production, decrease methane production, and reduce the accumulation of ammonia in the rumen [49]. These beneficial effects are largely attributed to modifications of the rumen microbial community, specifically the inhibition of methane-producing microorganisms such as methanogenic archaea [98]. Methanogenic archaea depend on the hydrogen produced during fermentation to convert carbon dioxide into methane. Table 3 shows the common methanogens and their reactions in the rumen. By limiting the availability of hydrogen, antibiotics restrict methane production by reducing the substrate required by methanogenic archaea to synthesise methane [99].

The use of antibiotics to mitigate methane production in ruminants has sparked a contentious debate owing to their various limitations and drawbacks. Firstly, antibiotics can encourage the development of antibiotic-resistant bacteria, which pose a serious threat to human and animal health [107]. Secondly, the potential presence of antibiotic residues in animal products, such as meat and milk, may represent a health concern for consumers [108]. Finally, the absence of stringent regulations governing antibiotic use in animal agriculture may lead to some farmers misusing antibiotics to increase feed efficiency or growth rates, rather than to reduce methane emissions [109]. Consequently, researchers have shifted their focus to identifying natural compounds with antimicrobial properties as alternatives to antibiotics. Plant extracts, such as saponins, tannins, and essential oils, have the ability to inhibit methanogenic bacteria in the rumen and decrease methane emissions [110,111]. Furthermore, some researchers have explored the feasibility of using seaweed extracts containing polysaccharides with antimicrobial properties as feed additives to limit the methane produced by ruminants [112]. 

Polysaccharides derived from both macro- and microalgae exhibit antimicrobial properties against yeasts and pathogenic bacteria. The mechanisms by which these polysaccharides exert their antimicrobial effects vary. The disruption of microbial cell walls is one of the predominant mechanisms employed by MAPs. MAPs bind to and disrupt the outer membranes of bacteria, causing the release of cellular components and, eventually, cell death (Figure 3). Sulphated galactans derived from *Eucheuma serra* and *Gracilari verrucosa* hindered the growth of *Escherichia coli* K88 by penetrating the cell wall and eventually reaching the interior of the bacterium. However, these sulphated galactans did not affect the proliferation of three intestinal probiotics or a yeast (*Saccharomyces cerevisiae*) [113]. The presence of sulphate or uronic acid groups in MAPs is closely linked to their antimicrobial properties. For example, κ-carrageenan, obtained from the red alga *Hypnea musciformis*, demonstrated effective antibacterial and antifungal effects against *Staphylococcus aureus* (IC_50_ = 48.2 μg/mL) and *Candida albicans* (IC_50_ = 147.3 μg/mL) [114]. Ulvan, extracted from the green seaweed *Ulva reticulata*, consists of a repeating disaccharide unit with a backbone of [→4)-D-GlcA (1 → 4)α-L-Rha3S-(1→]. It contains approximately 17.6% sulphate and 22.5% uronic acid and exhibits considerable antimicrobial activity against *Enterobacter cloacae* and *Escherichia coli* [115]. In contrast, the ulvan derived from *Ulva fasciata* is inactive against various bacteria, including *Bacillus cereus*, *Candida albicans*, *Escherichia coli*, *Micrococcus luteus*, *Pseudomonas aeruginosa*, and *Staphylococcus aureus* [116].

MAPs exhibit a remarkable ability to impede the attachment and colonisation of host cells by pathogens, thereby effectively preventing infection. Specific polysaccharides hinder the adhesion of pathogenic bacteria to intestinal cells, thereby inhibiting their colonisation and subsequent infection. The negatively charged sulphated groups within these polysaccharides are vital in interfering with pathogen adhesion to host cells. Pathogenic microorganisms rely on specific protein–carbohydrate interactions to bind to host cells and initiate infection. By mimicking these carbohydrate structures, the sulphated groups present in fucoidan and carrageenan can bind to the adhesion proteins of pathogens, obstructing their attachment to host cells [117]. This impedes the ability of the pathogen to colonise and infect the host. The results of an in vitro study demonstrated the effectiveness of fucoidans derived from *Fucus vesiculosus* and *Undaria pinnatifida* in a dose-dependent manner, disrupting the adherence of *Helicobacter pylori* to adenocarcinoma epithelial cells [118]. Additionally, sulphated galactans that were depolymerised and possessed a molecular weight of ≤20.0 kDa, obtained from *Eucheuma serra* and *Gracilaria verrucosa*, exhibited efficacy in inhibiting the adhesion of pathogenic bacteria, such as *Escherichia coli* K88, to the cell wall of *Saccharomyces cerevisiae* [119].

The immune-stimulating properties of MAPs provide an indirect mechanism through which to inhibit pathogens via promoting immune responses and triggering the production of reactive oxygen species (ROS). Algal polysaccharides can enhance the activity of immune cells, such as macrophages and natural killer cells, leading to the increased clearance of pathogens. Rabee et al. investigated the effects of a feed supplement containing a mixture of yeast (*Saccharomyces cerevisiae*) and microalgae (*Spirulina platensis* and *Chlorella vulgaris*) on feed intake, which resulted in enhanced immunological parameters in the blood of camels and sheep [120]. Additionally, the purification of sulphated polysaccharides from *Ulva pertusa* revealed that fractions with higher molecular weights demonstrated two-times-higher macrophage functionality compared with lower-molecular-weight fractions [121]. This highlights the importance of the structure–function relationship, particularly the impact of the molecular weight, on the biological activity of ulvan. Researchers have recently focused on harnessing ROS and oxidative stress to develop effective strategies against infection, based on the finding that microbicidal antibiotics induce ROS generation in host defence cells such as neutrophils and macrophages [122]. MAPs can also stimulate ROS production, which directly damages and eliminates pathogens. For example, sulphated polysaccharides derived from *Padina tetrastromatica* increased ROS generation [123]. 

In summary, the polysaccharides derived from marine algae hold considerable promise as substitutes for antibiotics and synthetic additives to reduce methane emissions from ruminants. These polysaccharides possess antimicrobial properties, are cost-effective, have low toxicity toward mammalian cells, and are readily available, so they could decrease both chemical use and drug resistance. In addition, these feed additives are sustainable and renewable, making them attractive solutions to reduce the environmental impact of animal agriculture while maintaining animal productivity and profitability. However, additional research is necessary to comprehensively assess their potential and optimise their use as feed additives in ruminant diets.

## 6. Future Trends and Conclusions

The use of MAPs as feed additives to mitigate methane emissions in ruminants presents a promising opportunity to address the pressing issue of greenhouse gas emissions. MAPs have demonstrated the ability to reduce methane production in ruminants through diverse mechanisms, including the modification of rumen fermentation, manipulation of microbial populations, modulation of gut-derived metabolites, and inhibition of the enzymes involved in methane synthesis. This critical review presented several notable findings regarding the effects of MAPs as feed additives for methane mitigation in ruminants. Firstly, the inclusion of MAPs in ruminant diets leads to the discernible modulation of the rumen microbial population, favouring the growth of beneficial microorganisms while inhibiting methanogens. This shift in microbial composition contributes to reduced methane production. Secondly, MAPs alter rumen fermentation patterns by promoting the production of propionate and other VFAs, while limiting the hydrogen available for methanogenesis. This metabolic redirection further contributes to decreased methane emissions. Finally, certain MAPs demonstrate antimicrobial activity, directly inhibiting methanogens and offering another avenue by which to reduce methane (Figure 4). 

Despite these positive outcomes, we identified limitations, such as the lack of long-term studies and the variability in the responses among different animal species and diets. To address these shortcomings, we recommend conducting more comprehensive, long-term studies. In addition, when assessing the effectiveness of these feed additives, we must consider their practical implementation and feasibility. Factors such as cost-effectiveness, scalability, the types of MAs, dosage, supplementation duration, and performance should be carefully evaluated to determine the viability of incorporating MAPs into livestock production systems. Furthermore, long-term studies and comprehensive assessments of animal health are essential to ensure the safety and well-being of ruminant livestock. Continued interdisciplinary research, collaboration, and innovation are increasingly important for the development of sustainable and effective strategies to reduce greenhouse gas emissions in the livestock industry. The integration of seaweed-based feed additives has substantial potential to promote a more sustainable and environmentally friendly approach to ruminant farming, thereby supporting global efforts to mitigate climate change.

## Figures and Tables

**Figure 1 polymers-15-03153-f001:**
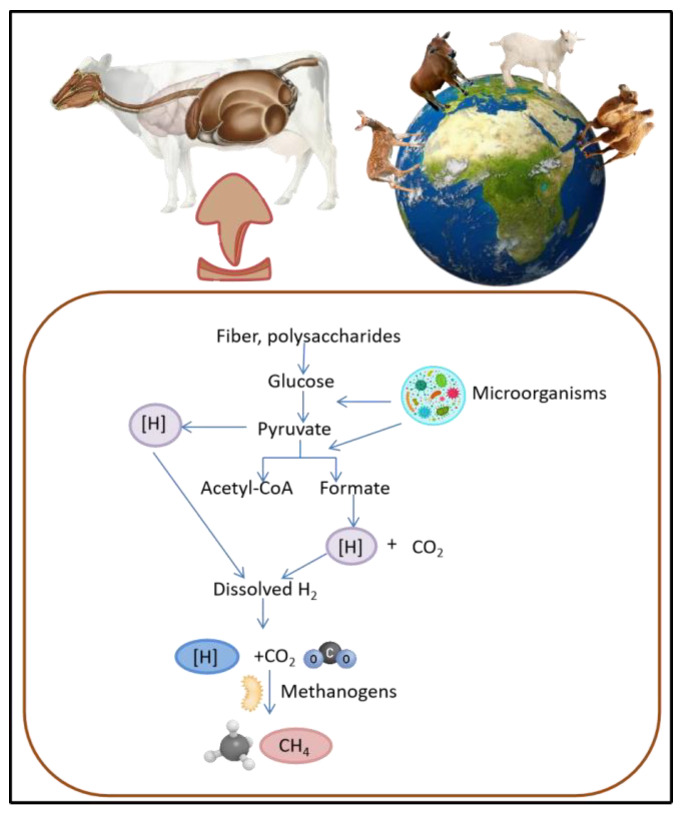
Pathway of methane emission in livestock. Methanogens use various organic compounds, including carbon dioxide and hydrogen, to produce methane gas as a byproduct, illustrating potential role of feed additives such as MAPs in mitigating methane emissions.

**Figure 2 polymers-15-03153-f002:**
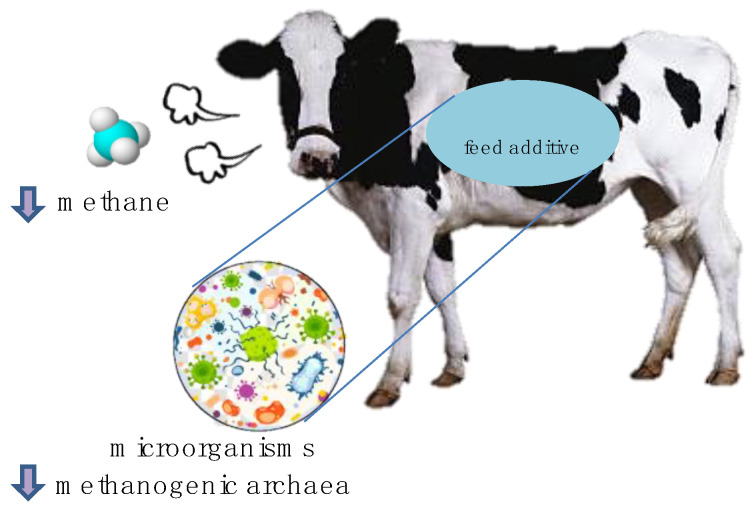
Feed additives (such as MAPs) reduce methane emissions from ruminant animals by regulating microorganisms, particularly by decreasing methanogenic archaea.

**Figure 3 polymers-15-03153-f003:**
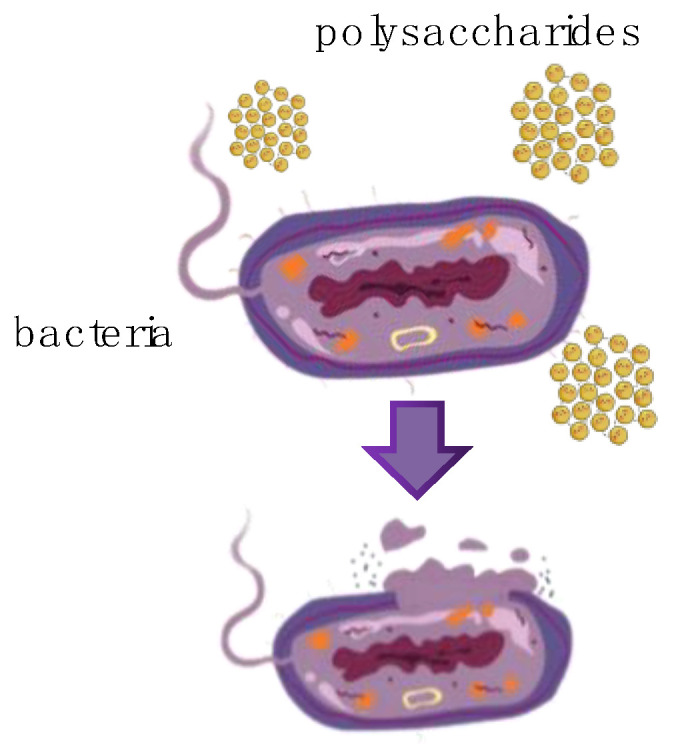
The antimicrobial mechanism of MAPs against bacteria involves increasing the permeability of bacterial cell membranes.

**Figure 4 polymers-15-03153-f004:**
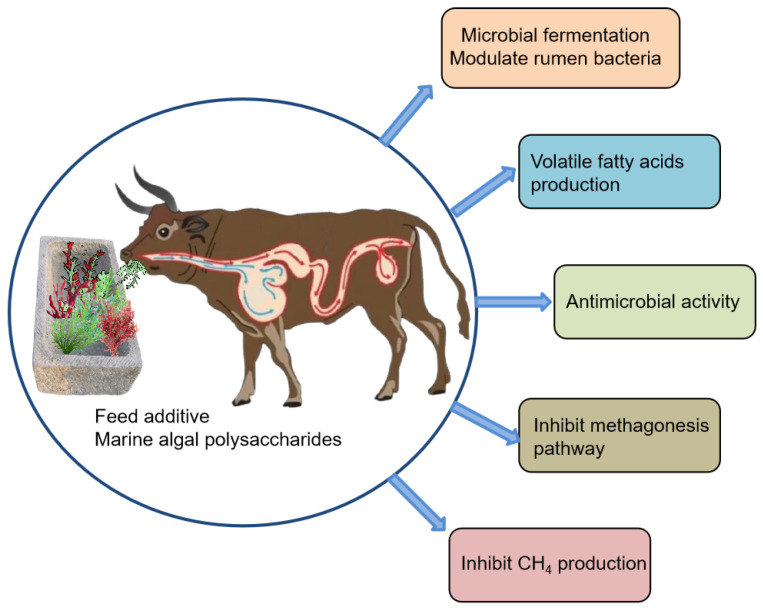
Application of marine algal polysaccharides as feed additives to reduce methane production.

**Table 1 polymers-15-03153-t001:** Estimates of methane reduction through the use of feed additives.

Feed Additive	Animal/In Vitro	Treatment	Methane Reduction (%)	Reference
3-Nitrooxypropanol	Cattle	10 mg/kg dry matter	39	[20]
Corn oil, wheat starch, marine algae	Dairy cows and goats	1.5% inclusion	28	[21]
*Asparagopsis armata*	Cows	1% inclusion level	47.2	[22]
origanum oil, hydrolysable tannins, and tea saponin	Sheep	40 mL/kg origanum oil	30	[23]
Grape marc	Dairy cows	5.0 kg dry matter of grape marc and 10.0 kg dry matter of ryegrass	15	[24]
*Nannochloropsis oceanica* (polysaccharide)	In vitro	2.5% incubation	10	[25]
*Macrocystis pyrifera* (polysaccharide)	In vitro	0.25 g of each diet	47.3	[26]
*Fucus vesiculosus* (polyphenol and polysaccharides)	In vitro	inclusion rate of 20% in dry matter	62.6	[27]
*Laminaria japonica*	In vitro	inclusion rate of 20% in dry matter	18.3	[28]
Sunflower and marine oils	In vitro	2.0% inclusion	16	[29]
*Ulva* sp. (ulvan)	In vitro	25% incubation	55	[30]
*Zonaria farlowii (*high starch and protein*)*	In vitro	5% inclusion	11	[31]

**Table 2 polymers-15-03153-t002:** Microorganisms involved in the volatile fatty acid process.

VFA	Microorganisms	Ref.
Acetic acid	*Acetobacter pasteurianus*, *A. aceti*, *Acetobacterium wieringae*, *Acetomicrobium flavidum*, *Acetobacterium woodii*, *Clostridium formicaceticum*, *C. aceticum*, *C. thermoaceticum*, *Gluconobacter strains*, *Moorella thermoacetica*, *Streptococcus lactis*, *Thermoanaerobacter kivui*	[67,68,69]
Propionic acid	*Propionibacterium freudenreichii*, *P. shermanii*, *P. acidipropionici*, *P. thoenii*, *P. jensenii*	[70]
Butyric acid	*Clostridium barkeri*, *C. thermobutyricum*, *C. butyricum*, *C. acetobutylicum*, *C. beijerinckii*, *Butyribacterium* sp., *Butyrivibrio fibrisolvens*, *Eubacterium*, *Fusobacterium nucleatum*, *Sarcinalimosum*, *Clostridium tyrobutyricum*	[71,72,73]
Isovaleric acid	*Propionibacterium freudenreichii*, *Pseudomonas* sp. strain VLB120	[74]

**Table 3 polymers-15-03153-t003:** The common methanogens and their reactions found in the rumen.

Feed Additive	Substrate	Reaction	Reference
*Methanobrevibacter gottschalkii*	Acetate Formate Pyruvate Methylamine Methanol Dimethylsulfide Acetate H_2_ CO CO_2_	H_2_ + CO_2_ → CH_4_ + 2H_2_O H_2_ + CH_3_OH → CH_4_ + H_2_O 4HCOO^−^ + 4H^+^ → CH_4_ + 3CO_2_ + 2H_2_O 4CO + 5H_2_O → CH_4_ + 3HCO_3_^−^ + 3H^+^ 4CH_3_OH → 3CH_4_ + HCO_3_^−^ + H_2_O + H^+^ 2(CH_3_)_2_S + 3H_2_O → 3CH_4_ + HCO_3_^−^ + 2H_2_S + H^+^ 4CH_3_NH_3_Cl + 2H_2_O → 3CH_4_ + CO_2_ + 4NH_4_Cl CH_3_COO^−^ + H_2_O → CH_4_ + HCO_3_^−^	[100]
*Methanobrevibacter millerae*			[101]
*Methanobrevibacter smithii*			[101]
*Methanobrevibacter thaueri*			[102]
*Methanobrevibacter ruminantium*			[75]
*Methanobrevibacter olleyae*			[103]
*Methanosphaera stadtmanae*			[104]
*Thermoplasmata*			[101]
*Methanomicrobium mobile*			[53]
*Methanobacterium lacus*			[53]
*Methanobacterium formicicum*			[105]
*Methanomicrobium bryantii*			[53]
*Methanosarcina barkeri*			[106]
*Methanosarcina mazei*			[53]

## Data Availability

Not applicable.

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
