# Peer review of "Role of Polysaccharides from Marine Seaweed as Feed Additives for Methane Mitigation in Ruminants: A Critical Review"

_polymers, 2023, doi:10.3390/polym15153153_

Round 1
Reviewer 1 Report
The manuscript presents the role of marine algal polysaccharides but there is a similar review which published in 2021in a better organization and more comprehensive way, Anim Nutr. 2021 Dec; 7(4): 1371–1387. So I would suggest the author add a bit more detail in the chemistry and effects on various aspects according to the methane mitigation. A better organization for the manuscript would be necessary as well.
More figures and tables would help to attract the interests of the readers. Such as A table to the part 5 presenting the previous studies would help to attract interests of the readers.
The author should add a discussion part in the end of the manuscript to discuss the findings, limitations, and recommendations.
There are also some minor suggestions.
In vitro and in vivo should be Italic. And the latin name should also be Italic, L357
references need to add to L49.
Add captions under the figures to describe and draw attention to important features.
Some references in the introduction part could be replaced by new published references.
Moderate editing of English language required
Some sentences have to check the grammar such as L344-346, L403
Author Response
- The manuscript presents the role of marine algal polysaccharides but there is a similar review which published in 2021in a better organization and more comprehensive way, Anim Nutr. 2021 Dec; 7(4): 1371–1387. So I would suggest the author add a bit more detail in the chemistry and effects on various aspects according to the methane mitigation. A better organization for the manuscript would be necessary as well.
Response: We sincerely appreciate your thorough assessment of our manuscript and the constructive suggestions you have provided. We acknowledge that the review published in 2021 focused on a broader range of marine algae compounds, including protein, amino acid, lipid, fatty acid, carbohydrate, and bromoform. In contrast, our review specifically centers on marine algae polysaccharides as feed additives for methane mitigation in ruminants. We understand the significance of this distinction and the need to emphasize the unique contribution of marine algae polysaccharides in our manuscript. To achieve this, we have included additional tables and figures to enhance reader understanding and highlight the specific role of polysaccharides in mitigating methane emissions. These visual aids will further support the clarity and coherence of our review, enhancing its value and relevance to the field of research.
- More figures and tables would help to attract the interests of the readers. Such as A table to the part 5 presenting the previous studies would help to attract interests of the readers.
Response: We sincerely appreciate your helpful suggestions. The table to the part 5 has been added.
- The author should add a discussion part in the end of the manuscript to discuss the findings, limitations, and recommendations.
Response: Thank you for your thoughtful comments and valuable input on our review.
We acknowledge its significance in providing a cohesive synthesis of the study's findings, limitations, and recommendations. We have added a dedicated the findings, limitations, and recommendations in the Future Trends part at the end of the manuscript to address these aspects.
- There are also some minor suggestions.
In vitro and in vivo should be Italic. And the latin name should also be Italic, L357
references need to add to L49.
Add captions under the figures to describe and draw attention to important features.
Some references in the introduction part could be replaced by new published references.
Response: We sincerely appreciate your meticulous attention to detail and valuable input. We have ensured the proper italicization of relevant terms throughout the manuscript to adhere to the required formatting. Furthermore, we have added descriptive captions under each figure to highlight essential features and draw attention to relevant information. Additionally, we have thoroughly reviewed the references in the introduction and replaced outdated ones with more recent and pertinent publications.
Reviewer 2 Report
The presentation of this manuscript was good enough in terms of presentation and methodology. Moreover, the paper is subjected to major improvement.
1. The aim and importance of this review is not clear.
2. How did the interaction between MAPs and rumen microbiota happen?
3. Compare the reduction of methane emission using different MAPs also compare them other feed additives. Better to do it at a table.
4. Give the detailed explanation and mechanism of the mitigation process in a sketch.
5. Is there any effect on meat and/or milk quality due to the use of MAPs?
Moderate editing of English language required
Author Response
The presentation of this manuscript was good enough in terms of presentation and methodology. Moreover, the paper is subjected to major improvement.
- The aim and importance of this review is not clear.
Response: Thank you for bringing this to our attention. To address this concern, we will revise the introduction section to provide a more concise and explicit statement of the aim and significance of the review.
- How did the interaction between MAPs and rumen microbiota happen?
Response: We sincerely thank the reviewer for their comment. Based on their valuable input, we have revised the review and included a discussion on the role of MAPs in promoting the growth of beneficial bacteria while inhibiting methanogens. These interactions have been found to contribute significantly to the reduction of methane emissions in ruminants.
- Compare the reduction of methane emission using different MAPs also compare them other feed additives. Better to do it at a table.
Response: We sincerely appreciate your helpful suggestions. The table has been added.
- Give the detailed explanation and mechanism of the mitigation process in a sketch.
Response: We appreciate reviewer’s suggestion for providing a detailed explanation and mechanism of the mitigation process in a sketch. As per reviewer’s recommendation, we have incorporate a visual representation to illustrate the mitigation process effectively.
- Is there any effect on meat and/or milk quality due to the use of MAPs?
Response:We sincerely thank the reviewer for their comment. We have revised and included a discussion in the review that examines the impact of MAPs on meat and milk quality.
Round 2
Reviewer 1 Report
The manuscript has been improve a lot on the quality and structure but there are still minor matters which could improve.
Add a full name before using the short abbreviation for DM.
Table 2, Acetomicrobium, should be Acetomicrobium sp. ?
Table 3, there are some feed additive without the substrate and reaction. Is it because of them are unknown? The author should add NA if the information is not available.
In the references, the journal name should use the abbreviation, such as Frontiers in Marine Science
It seems the author can move Figure 2 to Page 11 or 12.
The introduction should be shorten a bit, such as deleting L65, L107. A reference should add to L52.
The position of tables should be close to the related/cited texts.
Author Response
The manuscript has been improve a lot on the quality and structure but there are still minor matters which could improve. Add a full name before using the short abbreviation for DM.
Response: We sincerely appreciate your helpful suggestions. The abbreviations have been checked and revised.
Table 2, Acetomicrobium, should be Acetomicrobium sp. ?
Response: Thank you for your comments. The species name has been revised.
Table 3, there are some feed additive without the substrate and reaction. Is it because of them are unknown? The author should add NA if the information is not available.
Response: Thank you for bringing this to our attention. Since the substrates and reactions are commonly similar, we combine them in one column instead of corresponding to just one of them.
In the references, the journal name should use the abbreviation, such as Frontiers in Marine Science
Response: We sincerely appreciate your helpful suggestions. The abbreviation journal name has been revised.
It seems the author can move Figure 2 to Page 11 or 12.
Response: We sincerely appreciate your helpful suggestions. The Figure has been moved to suitable place.
The introduction should be shorten a bit, such as deleting L65, L107. A reference should add to L52.
Response: Thank you for bringing this to our attention. The sentences have been deleted and the reference has been added.
The position of tables should be close to the related/cited texts.
Response: Thank you for bringing this to our attention. The table has been revised.
Reviewer 2 Report
A table for comparison of reduction of methane emission using different MAPs is missing.
Minor editing of English language required
Author Response
A table for comparison of reduction of methane emission using different MAPs is missing.
Response: Thank you for bringing this to our attention. Table 1 provides a summary of the comparison of methane emission reduction using different chemical compounds and MAPs.